# Associations with the In-Hospital Survival Following Extracorporeal Membrane Oxygenation in Adult Acute Fulminant Myocarditis

**DOI:** 10.3390/jcm7110452

**Published:** 2018-11-20

**Authors:** Shaur-Zheng Chong, Chih-Yuan Fang, Hsiu-Yu Fang, Huang-Chung Chen, Chien-Jen Chen, Cheng-Hsu Yang, Chi-Ling Hang, Hon-Kan Yip, Chiung-Jen Wu, Wei-Chieh Lee

**Affiliations:** 1Division of Cardiology, Department of Internal Medicine, Kaohsiung Chang Gung Memorial Hospital, Chang Gung University College of Medicine, Kaohsiung 833, Taiwan; shauz@cgmh.org.tw (S.-Z.C.); cyfang@seed.net.tw (C.-Y.F.); ast42aiu@hotmail.com (H.-Y.F.); inq39@yahoo.com.tw (H.-C.C.); cjchen@cloud.cgmh.org.tw (C.-J.C.); yangch@cloud.cgmh.org.tw (C.-H.Y.); samuelhang@cloud.cgmh.org.tw (C.-L.H.); hkyip@cloud.cgmh.org.tw (H.-K.Y.); cvcjwu@cloud.cgmh.org.tw (C.-J.W.); 2Institute of Clinical Medicine, College of Medicine, National Cheng Kung University, Tainan 701, Taiwan; 3Division of Cardiology, Department of Internal Medicine, Chang Gung Memorial Hospital, Kaohsiung 123, Ta Pei Road, Niao Sung District, Kaohsiung City 83301, Taiwan

**Keywords:** acute fulminant myocarditis, extracorporeal membrane oxygenation, cardiogenic shock, decompensated heart failure, in-hospital mortality

## Abstract

Background: Acute fulminant myocarditis (AFM) is a serious disease that progresses rapidly, and leads to failing respiratory and circulatory systems. When medications fail to reverse the patient’s clinical course, extracorporeal membrane oxygenation (ECMO) is considered the most effective, supportive and adjunct strategy. In this paper we analyzed our experience in managing AFM with ECMO support. Methods: During October 2003 and February 2017, a total of 35 patients (≥18 years) were enrolled in the study. Twenty patients survived, and another 15 patients expired. General demographics, the hemodynamic condition, timing of ECMO intervention, and laboratory data were compared for the survival and non-survival groups. Univariate and multivariate Cox regression analyses were performed to identify the associations with in-hospital mortality following ECMO use in this situation. Results: The survival rate was 57.1% during the in-hospital period. The average age, gender, severity of the hemodynamic condition, and cardiac rhythm were similar between the survival and non-survival groups. Higher serum lactic acid (initial and 24 h later), higher peak cardiac biomarkers, higher incidence of acute kidney injury and the need for hemodialysis were noted in the non-survival group. Higher 24-h lactic acid levels and higher peak troponin-I levels were associated with in-hospital mortality. Conclusions: When ECMO was used for AFM, related cardiogenic shock and decompensated heart failure, higher peak serum troponin-I levels and 24-h serum lactic acid levels following ECMO use were independently associated with in-hospital mortality.

## 1. Introduction

Patients with acute myocarditis often present with a wide range of signs and symptoms including asymptomatic electrocardiographic modifications, chest pain, dyspnea or palpitations, and overt cardiac failure or sudden cardiac death, usually affecting children [1]. In adults, acute myocarditis is a rare and serious disease, and the burden of myocarditis as a percentage of prevalent heart failure (HF) varies from approximately 0.5% to 4.0% by age and region [2]. Acute myocarditis is defined by inflammation of the myocardium, with an inflammatory infiltrate with/without associated myocyte necrosis [3]. Acute fulminant myocarditis (AFM) is a rapidly progressive life-threatening disease accompanied by cardiogenic shock (CS) and decompensated HF; AFM sometimes requires mechanical circulatory support if conventional therapy cannot support the circulatory condition [4]. Previous studies have shown that mechanical circulatory support can save patients with CS resulting from AFM, and achieve an overall survival rate of 54.5–74.5% [5,6,7].

Factors influencing the prognosis of AFM are important, because some patients still need heart transplants for AFM-related HF to survive. Previous studies have reported that poor left ventricular systolic function and in-hospital arrhythmia were associated with poor short-term outcomes [8,9]. In children with AFM, initial serum troponin-I cutoff values greater than 14.21 ng/mL may also indicate the need for extracorporeal membrane oxygenation (ECMO) support [10]. However, most studies have focused on pediatric patients with AFM, and we hypothesized that the predictors of clinical outcomes would be similar in adult patients.

In adults, few studies have focused on the positive predictors of survival following ECMO use for AFM. Due to the limited availability of data on the clinical outcomes of ECMO use for adults with AFM, we investigated the clinical outcomes and explored the associations between clinical factors and in-hospital mortality in these patients.

## 2. Experimental Section

### 2.1. Patients and Groups

Between October 2003 and February 2017, adult patients (≥18 years) who experienced AFM and profound CS, and needed ECMO support were included. Peripheral percutaneous venoarterial (VA)-ECMO was used to stabilize the hemodynamic condition in patients who continued to experience unstable hemodynamics, refractory ventricular arrhythmia or asystole, or who required cardiopulmonary resuscitation (CPR). Patients were divided into the survival group or the non-survival group according to in-hospital mortality. In both groups, none of the patients received a left ventricular assist device (LVAD) and heart transplantation.

Coronary angiography was performed for all patients to exclude myocardial infarction if electrocardiography showed ST segment elevation and presented ventricular arrhythmia. During ECMO support, complete blood count/differential count, renal function, liver function, electrolyte, serum lactic acid, and arterial gas were followed up immediately and daily. An associated etiology survey of AFM was performed. General demographics, possible etiology, and mortality were collected for inpatients and outpatients from medical records and patient interviews. 

The Institutional Review Committee on Human Research at our institution approved this study protocol. All procedures followed were in accordance with the ethical standards of the responsible committee on human experimentation (institutional and national), and with the Helsinki Declaration of 1964 and later revisions.

### 2.2. Definitions

Profound CS was defined as systolic blood pressure (SBP) <90 mmHg for >30 min after correction of hypovolemia, hypoxemia, and acidosis under maximal medical treatment including vasopressors. Terminal disease was defined as medical history of malignancy with metastatic status, or advanced dementia, or severe chronic obstructive lung disease, or advanced heart failure, or decompensated liver cirrhosis, or stroke with bedridden status. All-cause mortality was defined as death from any cause.

### 2.3. Intra-Aortic Balloon Pumping (IABP), ECMO, and Distal Perfusion Device Criteria

IABP was set for patients with an unstable hemodynamic condition, including those whose SBP could not be maintained at >90 mmHg after intravenous administration of at least >20 μg/kg/min of dopamine. ECMO was considered for patients whose SBP could not be maintained at >75 mmHg after intravenous administration of at least two inotropic agents, or who experienced refractory ventricular arrhythmia or asystole, requiring CPR. However, patients were not placed on ECMO if they had a terminal disease, major bleeding problems (intracranial hemorrhage, and massive gastrointestinal bleeding), or fixed, dilated pupils without a light reflex post resuscitation. The need for a distal perfusion device was considered for patients who experienced no distal pulsation of their feet and cyanotic change of toes after ECMO implantation. 

Percutaneous femoral VA-ECMO cannulation was connected via femoral vessels (either unilateral or one side arterial, one side venous). The artery and vein were serially dilated, followed by placement of the arterial and venous ECMO cannulas. The venous cannula was either 50 cm in length and 21 Fr. (CAPIOX EBS Venous Cannula, Terumo, Japan); and the arterial cannula was 15 cm and 16.5 Fr. (CAPIOX EBS Arterial Cannula, Terumo, Japan). The cannulas were attached to the ECMO circuit that included an oxygenator and a Rotaflow centrifugal pump (CAPIOX SP Pump Console SP101, Terumo, Japan). The flow rate of the pump was initially set at approximately 2–3 L/min with the infusion of vasoactive agents to maintain a mean blood pressure ≥55 mm Hg, and the flow was then increased to a full scale (3.5–4.0 L/min) to maintain a mean blood pressure of ≥70 mm Hg.

Distal perfusion catheterization was performed as the proximal superficial femoral artery (SFA) ipsilateral to the arterial cannula was accessed antegrade using a micropuncture needle followed by a 0.018 in wire, and the 5 Fr. distal perfusion catheter was advanced into the SFA and attached to the side port of the arterial cannula with a three-way stopcock.

### 2.4. Study Endpoint

The study endpoint was all-cause mortality during the in-hospital period. 

### 2.5. Statistical Analysis

Data were expressed as numbers (percentages) and means ± standard deviation. Categorical variables were compared using Chi-square tests, and continuous variables were compared using *t*-tests. Univariate and multivariate Cox regression analyses were performed to identify the associations with in-hospital all-cause mortality and were expressed as hazard ratios (HRs) and 95% confidence intervals (CIs). Multivariate Cox regression analysis included all positive results of in-hospital all-cause mortality in univariate Cox regression analysis. Independent parameters of univariate Cox regression analysis were revised into multivariate Cox regression analysis. A Kaplan–Meier survival curve for all-cause mortality during the in-hospital period and the following one year was performed. All statistical analyses were performed using SPSS 22.0 (IBM Corp., Armonk, NY, USA), and *p* values <0.05 were considered statistically significant.

## 3. Results

### 3.1. Patients

During the study period, a total of 35 patients experienced AFM and needed ECMO placement for unstable hemodynamics or refractory ventricular arrhythmia or asystole, requiring CPR. During the in-hospital period, 20 patients survived and 15 patients died. The in-hospital survival rate was 57.1%. The in-hospital survival rate was similar during two different time periods (during October 2003 to December 2010: 58.8% (10/17) versus during January 2011 to February 2017: 55.6% (10/18); *p* = 1.000).

### 3.2. Comparisons of Baseline Characteristics between the Survival Group and Non-Survival Group

The average age in the survival group and non-survival group was 40.00 ± 14.73 years and 41.53 ± 14.97 years, respectively (*p* = 0.764). Forty-five percent of male presented in the survival group, and 60% male presented in the non-survival group (*p* = 0.500). The prevalence of diabetes mellitus, hypertension, chronic kidney disease, and thyroid disease was similar between the two groups. The presence of atrioventricular block or ventricular arrhythmia, and the percentage of ventilator use and IABP use were similar between the two groups (Table 1).

In the non-survival group, significantly higher initial and 24-h serum lactic acid levels, higher peak levels of cardiac biomarkers, and lower arterial pH values following ECMO use were noted. The parameters of cardiac echo were similar between the two groups, with poor left ventricular performance (LVP) in both. The incidence of acute kidney injury and the need for hemodialysis were higher in the non-survival group. The duration of shock to ECMO time, and the duration of ECMO use were similar between the two groups. The rate of complication was higher in the non-survival group, especially the incidence of ischemic legs, but not significantly different. 

### 3.3. Univariate and Multivariate Cox Regression Analyses for In-Hospital Mortality

Univariate Cox regression analysis identified that higher initial serum lactic acid, higher 24-h lactic acid, higher peak serum troponin-I, higher peak serum CK-MB, low PH value of arterial gas, the need for hemodialysis, and the incidence of hypoxic encephalopathy were significant factors for in-hospital mortality (Table 2).

Multivariate Cox analysis revealed that higher 24-h lactic acid levels and higher peak serum troponin-I levels were significant factors for in-hospital mortality.

### 3.4. Kaplan–Meier Survival Curve of In-Hospital Mortality

Overall, 57.1% (20/35) of patients survived during the in-hospital period. No patients experienced mortality after discharge during the one-year follow-up period (Figure 1).

### 3.5. The Reasons for All-Cause Mortality

Within the non-survival group, 7 patients died due to severe HF and multiple organ failure, 3 patients died due to intracranial hemorrhage, and another 5 patients died due to systemic inflammatory response syndrome and sepsis. 

## 4. Discussion

Myocarditis is an inflammatory disease of the myocardium caused by a broad range of pathogens including infectious, autoimmune, toxic, drug-induced, hypersensitive, or vasculitis [11]. Viral infection is the most prevalent etiology and is caused by enterovirus, adenovirus, parvovirus B19, human herpes virus 6, or cytomegalovirus [12]. Acute myocarditis is challenging for physicians because the presentation varies from asymptomatic electrocardiographic changes to fulminant HF and CS, and should be suspected if a young patient presents with unexplained cardiac abnormalities such as HF, cardiac arrhythmias, or conduction disturbances. Endomyocardial biopsy remains the gold standard technique for the diagnosis of myocarditis and inflammatory cardiomyopathy. However, this procedure requires an experienced operator and may bring some complications for patients in critical condition [13]. Therefore, an endomyocardial biopsy is recommended in unexplained new-onset HF of less than two weeks’ duration associated with hemodynamic compromise, or unexplained new-onset HF of two weeks to three months’ duration associated with a dilated left ventricle and new ventricular arrhythmias conduction disturbances, or a pseudo-infarct presentation after exclusion of coronary artery disease [14]. The diagnostic value of viral serology is limited, because of the patient’s satisfying general status, an endomyocardial biopsy is not necessary [12]. In our hospital, an endomyocardial biopsy is not routinely performed for AFM because of critical conditions, but is still considered if severe decompensated HF persists and heart transplantation needs to be considered. 

ECMO evolved from cardiopulmonary bypass, and provides prolonged hemodynamic and respiratory support outside of the operating suite [15]. With the development of technology and improvements in safety, the use of ECMO has expanded to patients with CS-related acute myocardial infarction, AFM, HF, acute circulatory failure attributable to intractable arrhythmias, and cardiac arrest [16]. ECMO has also been used for possible concomitant respiratory failure such as acute respiratory distress syndrome, symptomatic pulmonary hypertension, and refractory hypoxia [16]. According to work published by the European Society of Cardiology (2013), ECMO therapy is recommended and effective in rescuing patients with an unstable hemodynamic condition [17]. Several studies also prove that the use of ECMO could improve the short-term and long-term outcomes of AFM with CS and decompensated HF [7,17]. In our study, we also noted that patients could survive for one year if they could overcome an acute decompensated phase. 

The use of VA-ECMO raises the afterload of the left ventricle (LV), and could cause ultimately either further impairment or delay of cardiac contractility improvement. The rise in LV end-diastolic pressure and left atrial (LA) pressure also cause refractory severe pulmonary edema [18]. LA decompression shows good results in minimizing LA/LV volume/pressure overload, chamber dilation, and resolving severe pulmonary edema [19]. Several antegrade or retrograde methods of achieving LA decompression have been reported, including pulmonary artery drainage, atrial septostomy, transaortic pigtail LV drain, surgical LA/LV vent, and trans-septal cannula incorporated into ECMO [19,20]. IABP also indirectly decreases LV afterload and increases arterial diastolic pressure, concomitantly increasing coronary blood flow. However, the role of IABP is still controversial for CS, and no large studies have been reported for AFM related CS. In one meta-analysis study for CS, the combination strategy of IABP and ECMO did not influence mortality but was associated with lower mortality in comparison to patients on VA-ECMO alone in the patients with myocardial infarction [21]. Currently, no strong evidence of LV unloading has been reported in support of VA-ECMO, but some case series showed dramatic improvement [19,20]. In our study, LV venting showed no association with in-hospital mortality due to the small number of patients, and 80% (28/35) of patients received prior IABP setting. 

In addition, the use of ECMO should be instituted if patients with AFM do not recover from circulatory failure despite the use of cardiotonic or vasopressor drugs, and IABP treatment [7]. However, ECMO is not suitable for long-term support, and causes complications, including bleeding, hemolysis, leg ischemia, or multiple organ failure [15,22]. Even though the care of ECMO has improved, complications with ECMO are still common, and as expected, it is associated with a significant increase in morbidity and mortality [15]. Even though times have changed and there has been an improvement in medical care, the survival rate did not improve due to severe diseased status. Therefore, it is important to identify potential predictors of myocardial recovery in patients under ECMO support for AFM to define the subsequent decision-making algorithm in those who will not recover. Previous studies identified poor left ventricular systolic function and in-hospital arrhythmia to be associated with poor short-term outcomes in children with AFM. In adults with AFM, Nakamura et al. found that older age and complications related to ECMO were predictors of poor prognosis [22]. Some studies suggest that serum bilirubin levels might help in managing patients with AFM during ECMO; additionally, these studies suggest the need to consider converting peripheral ECMO to a ventricular assistant device in patients whose cardiac functions do not improve, and in those whose serum bilirubin levels have suddenly increased during ECMO support [23,24]. In our study, higher peak troponin-I levels, and higher 24-h lactic acidosis levels following ECMO use were independently associated with in-hospital mortality. Peak troponin-I indicated the amount of myocardium injury by pathogens, and 24-h lactic acid levels following ECMO use represented tissue perfusion status. Moreover, larger myocardium injury and poorer perfusion following ECMO use in patients with AFM predicted a poor short-term outcome. In addition, a higher 24-h lactic acidosis level also hinted that the patients had additionally poor leg perfusion after ECMO setting. In many countries, LVAD is still an expensive management, and heart transplantation is limited to obtaining the organ. According to the results of our study, the survey of LVAD implantation or urgent heart transplantation may be considered earlier in patients with higher peak troponin-I levels and higher 24-h lactic acidosis levels following ECMO use. 

The present study has several limitations; first, this was a retrospective single center study using observational analysis; second, this was a small sample size; third, no patients received LVAD or urgent heart transplantation; and fourth, even though we inserted IABP first for the study groups, some patients still needed the IABP to be removed due to the unsuitable arterial access for ECMO use (e.g. torturous vessel). In our country, LVAD is still an expensive device and the evaluation for heart transplantation requires a long time. Therefore, this study provides important hints about which AFM patients need to be considered for implantation of LVAD or urgent heart transplantation. Even though we only discussed the effects of ECMO use for patients who experienced AFM and CS, the clinical results are valuable because the number of adult patients with AFM following ECMO use was low. We still shared the precious experience about AFM with ECMO support and provided potential predictors of in-hospital mortality in adults with AFM and provided important information for healthcare in the future.

## 5. Conclusions

When ECMO was used for AFM-related CS and decompensated HF, higher peak serum troponin-I levels and 24-h serum lactic acid levels following ECMO use were independently.

## Figures and Tables

**Figure 1 jcm-07-00452-f001:**
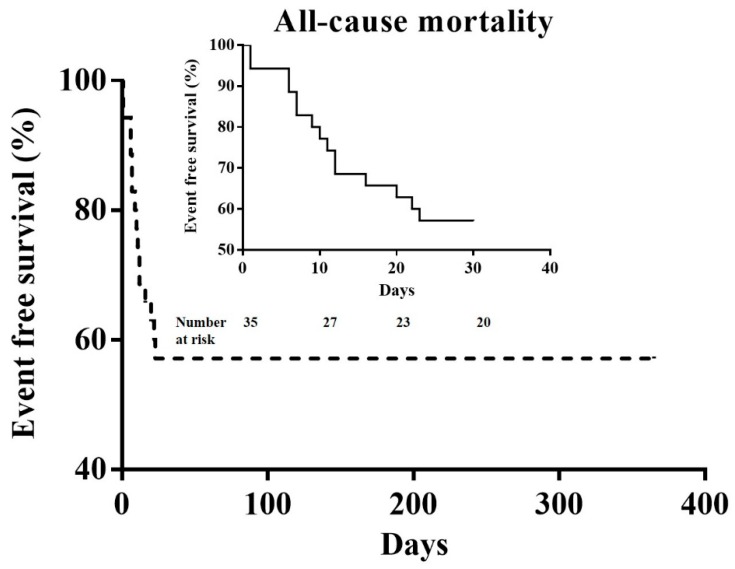
The Kaplan–Meier survival curve over a period of one-year for all-cause mortality. Analysis shows that 57.1% (20/35) of patients survived after the 30-day follow-up period. No patients experienced mortality after discharge during the one-year follow-up period.

**Table 1 jcm-07-00452-t001:** Baseline characteristics of study patients.

	Survival Group (*n* = 20)	Non-Survival Group (*n* = 15)	*p* Value
***General demographics***			
Age (years)	40.00 ± 14.73	41.53 ± 14.97	0.764
Male sex (%)	9 (45.0)	9 (60.0)	0.500
***Risk factors***			
Diabetes mellitus (%)	1 (5.0)	1 (6.7)	0.833
Hypertension (%)	2 (10.0)	1 (6.7)	0.727
Chronic kidney disease (%)	2 (10.0)	0 (0)	0.496
Thyroid dysfunction (%)	3 (15.0)	1 (6.7)	0.619
***The severity of hemodynamic condition***			
Out-hospital cardiac arrest (OHCA) (%)	2 (10.0)	4 (26.7)	0.367
CPR (%)	8 (40.0)	7 (46.7)	0.741
SBP (mmHg)	82.55 ± 34.23	67.33 ± 43.18	0.253
SBP < 80 mmHg (%) (except OHCA)	4 (22.2)	4 (36.4)	0.433
HR (beats/min)	95.10 ± 44.89	86.40 ± 53.58	0.605
HR > 120 beats/min (%) (except OHCA)	5 (27.8)	6 (50.0)	0.266
The number of inotropic agents use	0.700 ± 0.656	1.133 ± 0.516	0.043
***Cardiac rhythm***			
AV block	9 (45.0)	7 (46.7)	0.922
VT/VF	8 (50.0)	7 (46.7)	0.741
***Other supportive agents***			
IABP (%)	17 (85.0)	11 (73.3)	0.430
Ventilator (%)	14 (70.0)	14 (93.3)	0.199
Temporary pacemaker (%)	9 (45.0)	6 (40.0)	0.767
***Laboratory examination***			
White blood cell count (×10^3^)	14.11 ± 7.96	12.56 ± 5.94	0.543
Hemoglobin (g/dL)	12.59 ± 2.57	12.41 ± 2.90	0.853
Platelet (×10^3)^	205.18 ± 96.64	149.39 ± 84.60	0.095
BUN (mg/dL)	28.12 ± 17.56	29.93 ± 19.96	0.834
Creatinine (mg/dL)	1.80 ± 0.77	2.26 ± 0.47	0.437
AST (U/L)	1024.14 ± 794.24	1757.45 ± 722.92	0.517
ALT (U/L)	390.83 ± 156.49	829.46 ± 160.71	0.354
Total bilirubin (mg/dL)	1.15 ± 0.65	1.75 ± 1.05	0.089
CRP (mg/dL)	83.10 ± 47.01	82.84 ± 46.97	0.995
Lactate acid (initial) (mg/dL)	27.48 ± 11.53	67.59 ± 55.77	0.018
Lactate acid (24 h) (mg/dL)	16.93 ± 8.44	49.95 ± 33.38	0.001
Peak Troponin-I (ng/mL)	17.90 ± 6.03	62.87 ± 28.89	0.058
Peak CK-MB (ng/mL)	61.31 ± 29.78	233.40 ± 204.32	0.005
pH value of arterial gas	7.40 ± 0.11	7.20 ± 0.22	0.004
***Parameters of cardiac echo***			
LA dimension (mm)	31.38 ± 5.69	28.75 ± 5.08	0.218
LVESV (mL)	75.13 ± 32.01	73.67 ± 27.67	0.901
LVEDV (mL)	114.06 ± 45.66	103.67 ± 39.24	0.533
LVEF (%)	30.06 ± 12.19	27.42 ± 12.94	0.575
***The incidence of acute kidney injury (%)***	12 (60.0)	14 (93.3)	0.048
The need of hemodialysis (%)	4 (20.0)	11 (73.3)	0.002
***The incidence of hypoxic encephalopathy (%)***	2 (10.0)	5 (33.3)	0.112
***The duration of shock to ECMO use (minutes)***	182.06 ± 128.09	173.53 ± 138.89	0.858
***The duration of ECMO use (days)***	6.56 ± 4.26	9.50 ± 6.37	0.111
***The need of LV venting (%)***	2 (10.0)	2 (13.3)	0.759
***The complication of ECMO (%)***	9 (45.0)	11 (73.3)	0.167
Ischemic leg (%)	4 (20.0)	8 (53.3)	0.071
The need of distal perfusion device (%)	4 (20.0)	4 (26.7)	0.700
Bleeding (%)	7 (35.0)	7 (46.7)	0.511
ICH (%)	1 (5.0)	4 (26.7)	0.141

Data were expressed as mean ± SD or as percentage. Abbreviations: ECMO: Extracorporeal membrane oxygenation; CPR: Cardiopulmonary resuscitation; SBP: Systolic blood pressure; OHCA: Out-hospital cardiac arrest; HR: Heart rate; AV block: Atrioventricular block; VT: Ventricular tachycardia; VF: Ventricular fibrillation; IABP: Intra-aortic balloon pumping; BUN: Blood urea nitrogen; AST: Aspartate aminotransferase; ALT: Alanine aminotransferase; CRP: C-Reactive protein; CK-MB: Creatine kinase muscle/brain; LA: Left atrium; LVESV: Left ventricular end systolic volume; LVEDV: Left ventricular end diastolic volume; LVEF: Left ventricular ejection fraction; LV: Left ventricle; ICH: Intracranial hemorrhage.

**Table 2 jcm-07-00452-t002:** Univariate and multivariate Cox regression analyses for in-hospital all-cause mortality.

	Univariate Analyses			Multivariate Analyses		
Variables	Hazard Ratio	95% CI	*p* Value	Hazard Ratio	95% CI	*p* Value
Age > 60-year-old	2.060	0.271–15.675	0.485			
Male gender	1.730	0.614–4.874	0.299			
Shock to ECMO time (mins)	1.000	0.996–1.004	0.928			
OHCA	1.989	0.629–6.289	0.242			
CPR	1.154	0.418–3.185	0.783			
SBP < 80 mmHg	2.139	0.675–6.782	0.196			
HR > 120 beats/min	2.120	0.682–6.587	0.194			
Cardiac rhythm as VT/VF	1.154	0.418–3.185	0.783			
Platelet (×10^3^)	0.996	0.990–1.002	0.160			
Total bilirubin (mg/dL)	1.498	0.965–2.325	0.072			
Lactate acid (initial) (mg/dL)	1.020	1.006–1.033	0.003			
Lactate acid (24 h) (mg/dL)	1.057	1.029–1.086	<0.001	1.064	1.029–1.099	<0.001
Peak Troponin-I (ng/mL)	1.009	1.001–1.016	0.032	1.014	1.004–1.024	0.008
Peak CK-MB (ng/mL)	1.005	1.002–1.008	<0.001			
pH value of arterial gas	0.064	0.008–0.527	0.011			
LVEF (%)	0.988	0.944–1.033	0.592			
Acute kidney injury (%)	6.485	0.851–49.417	0.071			
The need of hemodialysis (%)	5.744	1.807–18.254	0.003			
Hypoxic encephalopathy (%)	3.623	1.220–10.763	0.020			
Ischemic leg (%)	2.389	0.864–6.607	0.093			
ICH (%)	2.853	0.110–1.119	0.077			
Prior IABP setting (%)	0.676	0.215–2.127	0.503			
The need of LV venting (%)	1.107	0.249–4.915	0.893			

Abbreviations: ECMO: Extracorporeal membrane oxygenation; OHCA: Out-hospital cardiac arrest; CPR: Cardiopulmonary resuscitation; SBP: Systolic blood pressure; HR: Heart rate; VT: Ventricular tachycardia; VF: Ventricular fibrillation; AST: Aspartate aminotransferase; ALT: Alanine aminotransferase; CK-MB: Creatine kinase muscle/brain; LVEF: Left ventricular ejection fraction; ICH: Intracranial hemorrhage; IABP: Intra-aortic balloon pumping; LV: Left ventricle.

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
