# Peer review of "Associations with the In-Hospital Survival Following Extracorporeal Membrane Oxygenation in Adult Acute Fulminant Myocarditis"

_jcm, 2018, doi:10.3390/jcm7110452_

Reviewer 1 Report

I thank the authors for the privilege to review the paper titled: `Associations with the In-hospital Survival Following  Extracorporeal Membrane Oxygenation in Adult  Acute Fulminant Myocarditis.´

This retrospective institutional database analysis is based on a good hypothesis. Still there seems to be some room for improvement:

Major:

First of all the authors have to define what they mean by pediatric (under 18 years, under 16 years, under 19 years..). Please do so in the abstract as well as in the Section Methods of the manuscript

It is obvious that more data should be revealed for readership. There is almost no descriptive statistics. Please include Tables with base line information of the cohort.

Please list reasons for death

The authors identified 24h serum lactic acid and Troponin I level as risk factors. For the 24h serum lactic acid it is important to know which patients had additional leg perfusion on ECMO (as I understood not all had additional leg perfusion). This might be one explanation.

The authors should comment how the left ventricle was unloaded. It is hard to believe that there was a sufficient unloading in all patients without venting the left sided chambers; and unloading is one of the important factors in patients suffering from acute myocarditis needing ECMO support.

It is very interesting why no patients underwent transplantation or durable VAD implantation. Could you comment on that?

Minor points:

Text formatting needed page 3 (lines 101-102), page 5 lines 164 – 175

Please add on page 2 line 70: CA was performed IN ALL PATIENTS to exclude myocardial infarction

Author Response

Responses to the reviewer 1 comments

Dear Reviewer 1:

   Your constructive criticism is greatly appreciated. We have made the following responses to comply with your honorable suggestions:

Comments

Point 1:

First of all the authors have to define what they mean by pediatric (under 18 years, under 16 years, under 19 years.). Please do so in the abstract as well as in the Section Methods of the manuscript

Response 1:

Dear reviewer: Thanks for your kindly suggestion. We added the definition of adult, which was ≥ 18 years in the abstract and the method of manuscript.

Point 2:

It is obvious that more data should be revealed for readership. There is almost no descriptive statistics. Please include Tables with base line information of the cohort.

Response 2:

Dear reviewer: Thanks for your kindly suggestion. We added the table 1 and table 2 in the manuscript.

Point 3:

Please list reasons for death

Response 3:

Dear reviewer: Thanks for your kindly suggestion. We added one paragraph in results, in page 7, line 175-178, as “Among non-survival group, 7 patients died due to severe HF and multiple organ failure, 3 patients died due to intracranial hemorrhage and another 5 patients died due to systemic inflammatory response syndrome and sepsis”.

Point 4:

The authors identified 24h serum lactic acid and Troponin I level as risk factors. For the 24h serum lactic acid it is important to know which patients additional leg perfusion on ECMO had (as I understood not all had additional leg perfusion). This might be one explanation.

Response 4:

Dear reviewer: Thanks for your kindly suggestion. We added one paragraph in discussion, in page 8, line 245-247, as “In addition, a higher 24-hour lactic acidosis level also hints the patients had additionally poor leg perfusion after ECMO setting”.

Point 5:

The authors should comment how the left ventricle was unloaded. It is hard to believe that there was a sufficient unloading in all patients without venting the left sided chambers; and unloading is one of the important factors in patients suffering from acute myocarditis needing ECMO support.

Response 5:

Dear reviewer: Thanks for your kindly suggestion. We added one paragraph in discussion, in page 7 and 8, in line 209-224, as “The use of VA-ECMO rise the afterload of the left ventricle (LV) and could cause ultimately either further impairment or delay of cardiac contractility improvement. The rising LV end-diastolic pressure and left atrial (LA) pressure also cause refractory severe pulmonary edema. [19] LA decompression shows good results in minimizing LA/LV volume/pressure overload, chamber dilation, and resolving severe pulmonary edema. [20] Several antegrade or retrograde methods of achieving LA decompression have been reported, including pulmonary artery drainage, atrial septostomy, transaortic pigtail LV drain, surgical LA/LV vent and trans-septal cannula incorporated into ECMO. [20, 21] IABP also indirectly decrease LV afterload and increase arterial diastolic pressure, concomitantly increasing coronary blood flow. However, the role of IABP is still controversial for CS and no large studies were reported for AFM related CS. In one meta-analysis study for CS, the combination strategy of IABP and ECMO did not influence mortality but was associated with lower mortality in comparison to patients on VA-ECMO alone in the patients with myocardial infarction. [22] Currently, no strong evidence of LV unloading was reported in VA-ECMO support, but some case series showed dramatic improvement. [20, 21] In our study, LV venting showed no associated with in-hospital mortality due to small number of patients and 80 % (28/35) patients received prior IABP setting”. We also added associated references.

Point 6:

It is very interesting why no patients underwent transplantation or durable VAD implantation. Could you comment on that?

Response 6:

Dear reviewer: Thanks for your kindly suggestion. We also discussed this issue in the limitation in page 8, in line 253-258, as “Third, no patients received LVAD or urgent heart transplantation. ……. In our country, LVAD is still an expensive device and the evaluation of heart transplantation need long time. Therefore, this study provided the important hints about which AFM patients need to be considered implantation of LVAD or urgent heart transplantation”. In the future, the implantation of LVAD will be included in our healthcare insurance in my country. Therefore, we could provide an important information for healthcare of my government about the need of LVAD or urgent heart transplantation for these critical AFM patients. In addition, we also discussed this in limitation, in page 8, in line 258-262, as “only discussed the effect of ECMO use for patients who experienced AFM and CS, but the clinical results are valuable because the number of adult patients with AFM following ECMO use was few. We still shared the precious experience about AFM with ECMO support and provided potential predictors of in-hospital mortality in adults with AFM and provided important information for healthcare in the future”.

Point 7:

Text formatting needed page 3 (lines 101-102), page 5 lines 164 – 175.

Response 7:

Dear reviewer: Thanks for your kindly suggestion. We corrected the format of text.

Point 8:

Please add on page 2 line 70: CA was performed IN ALL PATIENTS to exclude myocardial infarction

Response 8:

Dear reviewer: Thanks for your kindly suggestion. We modified one paragraph in page 2, line 70, as “Coronary angiography was performed in all patients to exclude myocardial infarction if electrocardiography showed ST segment elevation and presented ventricular arrhythmia”.

Thank you, very, very much for your kind help!

Reviewer 2 Report

The paper Chong and coll. Is aimed to report the parameters associated with a worse prognosis in adult patients undergone ECMO for acute fulminant myocarditis. The authors enrolled a small sample over a long period of observation. The following point should be addressed:

-    Was the enrollment data of survival group different from that of non survival group? I expect that for a learning curve patients implanted more recently were characterized by a greater life expectancy.

- How many patients were candidate to urgent heart transplantation and or to the implantation of left ventricular assist device? How many patients did it?

- The cause of death should be reported.

- Table 1 and 2 should not be in the supplementary material but they should be included in the text.

Author Response

Responses to the reviewer 2 comments

Dear Reviewer 2:

   Your constructive criticism is greatly appreciated. We have made the following responses to comply with your honorable suggestions:

 Comments

Point 1:

Was the enrollment data of survival group different from that of non survival group? I expect that for a learning curve patients implanted more recently were characterized by a greater life expectancy.

Response 1:

Dear reviewer: Thanks for your kindly suggestion. We added one paragraph in results, in page 3, line 126-128, as “The in-hospital survival rate was similar between two different time period (during October 2003 to December 2010: 58.8 % (10/17) vs. during January 2011 to February 2017: 55.6 % (10/18); p = 1.000)” and in discussion, in page 8, line 230-231, as “Even though the change of times and the improvement of medical care, the survival rate did not improve due to severe diseased status”.

Point 2:

How many patients were candidate to urgent heart transplantation and or to the implantation of left ventricular assist device? How many patients did it?

Response 2:

Dear reviewer: Thanks for your kindly suggestion. Actually, the survey of heart transplantation will be started, and the patient will be a candidate for urgent heart transplantation if the patient presents severe heart failure status and needs mechanical support. However, LVAD is still an expensive management and heart transplantation is limited to organ obtain in my country. Therefore, no patients received urgent heart transplantation or the implantation of LVAD during study period. In the future, the implantation of LVAD will be included in our healthcare insurance in my country. According to the results of our study, the survey of implantation of LVAD or urgent heart transplantation may be considered earlier in the patients with higher peak troponin-I level and higher 24-hour lactic acidosis level following ECMO use. We could provide an important information for healthcare of my government about the need of LVAD or urgent heart transplantation for these critical AFM patients. We also discussed this issue in the limitation in page 8, in line 253-258, as “Third, no patients received LVAD or urgent heart transplantation. ……. In our country, LVAD is still an expensive device and the evaluation of heart transplantation need long time. Therefore, this study provided the important hints about which AFM patients need to be considered implantation of LVAD or urgent heart transplantation”.

Point 3:

The cause of death should be reported.

Response 3:

Dear reviewer: Thanks for your kindly suggestion. We added one paragraph in results, in page 7, line 175-178, as “Among non-survival group, 7 patients died due to severe HF and multiple organ failure, 3 patients died due to intracranial hemorrhage and another 5 patients died due to systemic inflammatory response syndrome and sepsis”.

Point 4:

Table 1 and 2 should not be in the supplementary material but they should be included in the text.

Response 4:

Dear reviewer: Thanks for your kindly suggestion. We added the table 1 and table 2 in the manuscript.

Thank you, very, very much for your kind help!
